# The Recent Management of Vestibular Schwannoma Radiotherapy: A Narrative Review of the Literature

**DOI:** 10.3390/jcm13061611

**Published:** 2024-03-11

**Authors:** Lucie Brun, Thierry Mom, Florent Guillemin, Mathilde Puechmaille, Toufic Khalil, Julian Biau

**Affiliations:** 1Department of Radiation Oncology, University Hospital of Lyon, 69002 Lyon, France; lucie.brun@chu-lyon.fr; 2Department of Otolaryngology-Head and Neck Surgery, University Hospital of Clermont-Ferrand, 63000 Clermont-Ferrand, France; tmom@chu-clermontferrand.fr (T.M.); mpuechmaille@chu-clermontferrand.fr (M.P.); 3Department of Radiation Oncology, Jean Perrin Center, 51 rue Montalembert, 63100 Clermont-Ferrand, France; florent.guillemin@clermont.unicancer.fr; 4Department of Neurological Surgery, University Hospital of Clermont-Ferrand, 63000 Clermont-Ferrand, France; tkhalil@chu-clermontferrand.fr; 5INSERM U1240 IMoST, University of Clermont Auvergne, 63000 Clermont-Ferrand, France

**Keywords:** radiotherapy, radiosurgery, vestibular schwannoma

## Abstract

Background: Radiotherapy (RT) plays an important role in the therapeutic management of vestibular schwannoma (VS). Fractionated stereotactic radiotherapy (FSRT) or radiosurgery (SRS) are the two modalities available. The purpose of this article is to review the results of VS RT studies carried out over the last ten years. Materials and Methods: A literature search was performed with PubMed and Medline by using the words vestibular schwannoma, acoustic neuroma, radiotherapy, and radiosurgery. Results: In small (<3 cm) VS, SRS offers a local control rate of >90%, which seems similar to microsurgery, with a favorable tolerance profile. Hypofractionated FSRT (three to five fractions) is a relatively recent modality and has shown similar outcomes to normofractionated FSRT. Hearing preservation may highly differ between studies, but it is around 65% at 5 years. Conclusions: SRS and FRST are non-invasive treatment options for VS. SRS is often preferred for small lesions less than 3 cm, and FSRT for larger lesions. However, no randomized study has compared these modalities.

## 1. Introduction

Vestibular schwannoma (VS) is the most common tumor of the cerebellopontine angle [1]. The incidence is estimated at around 3–5 per 100,000 person-years [2]. VS is a benign, slow-growing nerve sheath tumor. VS natural history suggests that the growth rate is variable and is around 2.9 +/− 1.2 mm per year. Common symptoms include hearing impairment (in 90% of cases), tinnitus, and vestibular disorders, which can impair the patient’s quality of life.

Given the anatomic relationship of VS, tumor growth can lead to facial nerve (FN) dysfunction, trigeminal nerve dysfunction (TN), cerebellar dysfunction, brainstem compression, and hydrocephalus [3].

In the absence of randomized trials, the management of VS is based on several factors, such as the size of the VS, the symptoms, the comorbidities of the patient, and the patient’s preferences. The management is decided by a multidisciplinary tumor board among three options: wait and see, surgery, and radiation therapy (RT) [4,5,6,7,8].

RT techniques employed to treat these patients include stereotactic radiosurgery (SRS) (using various treatment systems such as Gamma Knife [GK], CyberKnife [CK], or dedicated linear accelerator [LINAC]) or fractionated stereotactic radiotherapy (FSRT), and all these methods have the goal of arresting tumor growth while minimizing the risk to adjacent structures.

This article aims to compile data (efficacy and toxicity) on the various RT techniques to treat VS over the last ten years in order to guide the choice of treatment strategy.

## 2. Materials and Methods

A comprehensive literature search was conducted across multiple databases, including PubMed and MEDLINE. The search aimed to capture a wide spectrum of studies on VS and their associated treatment approaches. Key terms used for this search encompassed both specific and general terms such as “acoustic neuroma”, “vestibular schwannoma”, “radiotherapy”, and “radiosurgery ”. Boolean operators, AND and OR, were employed to combine these terms for a comprehensive search.

The period was from 2013 to 2023. The relevance was assessed based on the number of patients, the length of follow-up, the completeness of analysis of RT technical data, outcomes in terms of local control, and early and late side effects.

A narrative synthesis was conducted, categorizing findings based on the type of management approach.

## 3. Results

We found 451 articles, and 30 were selected for the analysis.

### 3.1. Stereotactic Radiosurgery (SRS, Table 1)

SRS utilizes multiple convergent beams to deliver a high single dose of radiation to a radiographically defined treatment volume, thereby minimizing dose delivery to adjacent structures. GK, CK, and LINAC are the most common machines used for SRS. SRS is an appropriate technique for tumors up to approximately 3 cm in diameter (KOOS I, KOOS II), with well-established long-term outcomes. 

Although no randomized trials have compared SRS with microsurgery, numerous studies suggest that SRS achieves similar local control to that of surgery, with rates of hearing preservation that seem more favorable [9,10,11,12,13,14]. 

In most series, tumor control is reported between 90% and 99%, whereas cranial nerve complications such as facial nerve deficit and trigeminal neuropathy are reported in ranges of 0% to 5% and 1% to 21%, respectively [5,15,16,17]. A large study published in 2019 [18] reported the outcomes of 1002 VS treated from 2005 to 2018 with GK SRS. The median follow-up was 3.6 years. Local control at 3, 5, and 10 years was 96.6%, 92.3%, and 90.8%, respectively. A total of 3% of patients had trigeminal sensory dysfunction, and 0.9% had a permanent facial weakness. 

**Table 1 jcm-13-01611-t001:** Patient tumor and outcomes data for Stereotactic radiosurgery.

	Author	Year	Number of Patients		Median Follow-Up (Months)	Median Dose (Gy)	Median Volume (cc)	Local Control	Treatment after Radiotherapy	Toxicity			
			Total	Treatment Before						Hearing Preservation	Trigeminal Nerve Dysfunction	Facial Nerve Dysfunction	Other
SRS	Rueb	2017	49	4 post-surgery	65	12.6	0.24	100%	0	78%	0%	2% (grade 1)	NA
	Windisch	2019	996	175 post-surgery	42	13	0.61	96.6% at 3 years 92.3% at 5 years 90.8% at 10 years	13 SRS, 16 surgeries	21.50%	3%	0.90%	0.5% hydrocephalus requiring shunt
	Jonhson	2019	871	NA	62	13	0.9	97% at 3 years 95% at 5 years 94% at 10 years	11 surgeries, 6 SRS	76.9% at 3 years 68.4% at 5 years 51.4% at 10 years	5.80%	1.60%	1.7% CSF DIVERSION
	Rueb	2018	335	70 post-surgery	30	13	1.1	98% at 2 years 89% at 5 years 88% at 10 years	12 surgeries, 8 SRS	56% at 5 years	3.80%	3.60%	0.8% hydrocephalus with one required shunt
	Klijn	2016	420	NA	60	11	1.4	91.3% at 5 years 84.8% at 10 years	NA	65% at 3 years 42% at 5 years	3.10%	1%	1.2% hydrocephalus
	Yeole	2021	34	8 post-surgery	34.7	12	10.9	85.30%	5 surgeries	0%	12.50%	4%	NA
	Mezey	2020	103	22 post-surgery	74	12.5	13.6	78.60%	17 surgeries	25.20%	8.70%	2.90%	14.6% hydrocephalus
	Park	2023	106	0	148	12.5	3.68	95.3% at 3 years 94.3% at 5 years 87.7% at 10 years 86.6% at 15 years	NA	46.40%	4.70%	2.80%	4.9% hydrocephalus
	Tatagiba	2023	559	NA	82.2	12–13 Gy		89%	37	54%	NA	NA	NA

In recent years, the challenge has been long-term hearing preservation, which remains one of the major concerns of SRS. Johnson et al. [15] analyzed the results of 871 patients treated with GK SRS. The median follow-up was 5.2 years. They reported a progression-free survival after SRS of 97% at 3 years, 95% at 5 years, and 94% at 10 years. They reported a serviceable hearing preservation rate of 68.4% at 5 years.

A recent meta-analysis reported the outcomes of 1409 patients [19]. Tumor control was achieved in 96.1%. Hearing preservation was found in 59.4%, with a median follow-up of 6.7 years. The main favorable prognostic factors were young age, good hearing status at SRS, early treatment after diagnosis, small tumor volume, low marginal irradiation dose (<13 Gy), and maximal dose to the cochlea.

A study analyzing tumor control rates with SRS in relation to pretreatment growth rates found that patients with pretreatment growth rates of less than 2.5 mm/year had control rates of 97% at an average of 43.5 months post-treatment, while those with pretreatment growth rates greater than that had tumor control rates of 69% in the same follow-up period [19]. 

Tumor volume has been reported in many studies as a major prognostic factor of local control. Rueb et al. [20], with a median volume of 0.24 cc in 45 patients receiving an average marginal dose of 12.6 Gy, found a radiological tumor control of 100% and preserved serviceable hearing at 78%. Turek et al. [21], with a median volume of 0.44 cc in 621 patients, found a 96.8% local control, with a hearing preservation rate of 60.8%. For larger VS, a recent study of 2020 [22] involving 103 patients with a mean follow-up of 6.2 years and a mean volume of 13.6 cc found a local control of 78.6%, with a 14.6% risk of hydrocephalus requiring therapeutic intervention. Yeole et al. [23], with a mean tumor volume of 10.9 cc, reported more encouraging results, with disease control in 94.2% of cases, although 14.7% of patients required post-SRS surgical management. These studies confirm that large VS are associated with higher morbidity rates and a reduced tumor control rate after SRS [24].

### 3.2. Fractionated Stereotactic Radiotherapy (FSRT) (Table 2)

FFSRT provides an attractive management strategy because the lower dose per fraction allows for the treatment of larger lesions. The temporally spaced treatment increases the probability of targeting tumor cells during periods of high radiosensitivity and cell division, as well as permitting better normal tissue sparing. Furthermore, this lower dose per fraction is thought to minimize hearing loss and cranial nerve damage. The dose can be delivered using conventional fractionation (NormoFSRT) (daily dose of 1.8 to 2 Gy to a total dose of 50 to 54 Gy) or hypofractionation (HypoFSRT) (3 to 7 Gy per fraction to a total dose of 21 to 35 Gy).

**Table 2 jcm-13-01611-t002:** Patient tumor and outcomes data for fractionated stereotactic radiotherapy.

	Author	Year	Number of Patients		Median Follow-Up (Months)	Median Dose (Gy)	Median Volume (cc)	Local Control	Treatment after Radiotherapy	Toxicity			
			Total	Treatment Before						Hearing Preservation	Trigeminal Nerve Dysfunction	Facial Nerve Dysfunction	Other
HypoFSRT	Gawish	2023	134	NA	54	22 Gy in 5 fractions	1.65	96% at 3 years 95% at 5 years 94% at 7 years	4 surgeries, 2 SRS	60%	0%	1.50%	NA
	Pialat	2021	82	16	48	21 Gy in 3 fractions or 25 Gy in 5 fractions	1.01	96.30%	3 surgeries	46%	3.60%	2.40%	NA
	Patel	2017	383	NA	72	25 Gy in 5 fractions		98%	9	36.20%	0.30%	3.90%	1.3% hydrocephalus
normoFSRT	Litre	2013	155	13 surgeries	60	50.4 Gy in 28 fractions	2.45	99.3% at 3 years 97.5% at 5 years 95.2% at 7 years	NA	54%	3.20%	2.50%	3.8% hydrocephalus
	Champ	2013	154	NA	35	46.8 Gy in 26 fractions	2.41	99% at 3 years 93% at 5 years	NA	64% at 3 years 54% at 5 years	1.90%	1.90%	1.9% hydrocephalus
	Barnes	2018	95	14 surgeries	51	54 Gy in 30 fractions		97%	4 surgeries, 2 SRS	43%	NA	2.10%	NA
					88	50.4 in 28 fractions		95%		64%	NA		NA
					78	59.4 Gy		92%			NA		NA
	Saraf	2022	20	0	48	50.4–54 Gy in 28–30 fractions	0.81	100%	0	53%	0.00%	NA	NA

#### 3.2.1. NormoFSRT

NormoFSRT was initially studied as part of adjuvant treatment for non-radically removed VS [25] and then became an alternative for the treatment of VS not eligible for microsurgery. This treatment has been studied in numerous historical studies, but few recent studies have focused on the subject. In 2014 [26], a review listed seven studies from the 2000s delivering doses between 50 Gy and 57 Gy. Local control rates ranged from 91.4% to 100%, hearing preservation rates ranged from 71.4% to 98%, and only a few other cranial nerve toxicities were reported (<4%). Woolf et al. [27] reported in 2013 the outcomes of 93 patients with a median follow-up of 5.7 years (0.8–15.3). The median total dose was 52.5 Gy in 25 fractions. Local control was 92%. Hearing preservation was 93%, and less than 1% had cranial nerve toxicities. In 2013, another study [28] reported the outcomes of 158 patients treated by FSRT between January 1996 and December 2009. Patients received a total dose of 50.4 Gy in five weekly fractions of 1.8 Gy weekly. Median tumor volume was 2.45 cc (range, 0.17–12.5 cc). Local tumor control rates were, respectively, 99.3%, 97.5%, and 95.2% at 3-, 5-, and more than 7-year follow-up. Neurological sequelae consisted of radiation-induced trigeminal nerve impairments (3.2%), facial neuropathies (2.5%), and new or aggravated tinnitus (2.1%). Champ et al. [29] treated 154 patients at 46.8 Gy in 26 fractions of 1.8 Gy. They found local control rates of 99% and 93% at 3 and 5 years, respectively. The preservation of hearing function was estimated at 66% and 54% at 3 and 5 years, respectively. These results support dose de-escalation while maintaining excellent local control.

#### 3.2.2. HypoFSRT

In contrast, hypoFSRT has been extensively studied over the last ten years. HypoFSRT has been favored over conventional normoFSRT because of less frequent treatment visits, comparable tumor control, and equally low complication risk. The hypothesis of using HypoFRST over SRS is that this would maintain the excellent tumor control achieved with high daily dose treatment but with reduced toxicities while keeping the radiobiological advantage of fractionation. 

Several studies investigated three- and five-fraction hypoFSRT delivering between 21 and 25 Gy. One of the largest experiences was conducted at the University of Baltimore [30], with a median follow-up of 72 months: 383 patients were treated from 1995 to 2007 with hypoFSRT (25 Gy in five fractions). Local control was achieved in 78%. Nine patients experienced treatment failure requiring salvage microsurgery. Preservation of serviceable hearing was achieved in 51% at lost follow-up. In this cohort, 44% of long-term responders had a transient tumor progression after HSRT. In a recent retrospective study, Gawish et al. [31] analyzed 134 patients treated with hypoFSRT for VS with a total dose of 22 Gy in five fractions at 95% isodose line. Local control at 3, 5, and 7 years was 96%, 95%, and 94%, respectively. Four patients underwent surgical intervention after hypoFSRT. Serviceable hearing preservation was obtained in 90.5%.

Pialat et al. [32] reported the outcomes of 82 patients treated with CK hypoFSRT with a total dose of 3 × 7 Gy or 5 × 5 Gy. They showed 95% local control at 5 years. Three patients required surgery after hypoFSRT. Hearing deterioration was observed in 46%, and a facial nerve complication rate (paralysis, spasm) of less than 10% was reported. Univariate factors associated with relapse were a GTV above 2 cc and a dosimetry compliance index below 1.1.

#### 3.2.3. Proton Radiation Therapy

Proton RT is a form of heavy particle RT used in a limited number of specialized centers around the world. Protons achieve greater avoidance of normal tissue radiation dose than photon-based techniques based on the characteristic of the proton beam, which deposes energy at the end of a linear track, beyond which there is a rapid fall-off in dose. In a prospective study [33], 20 patients received between 50.4 and 54 Gy proton RT from 2010 to 2019. They showed a local control rate of 100% at 1 year but did not meet the primary endpoint with a serviceable hearing preservation of 53%. Barnes et al. [34] studied proton RT dose levels of 50.4 Gy to 54 Gy. The local control rate was over 95%, and hearing preservation was 43% in the 54 Gy group and 64% in the 50.4 Gy group, with no statistical difference. Finally, in his review, Santacroce et al. [35] described that proton RT did not seem to offer an advantage for facial and hearing preservation compared to most of the currently reported SRS series. 

### 3.3. Comparison between SRS and FRST (Table 3)

No randomized trial has directly compared SRS and FSRT. The results are often difficult to interpret since the choice of the technique in many studies is often made based on tumor size, which is known to be a prognostic factor of local control. This was the case in the study by Huo et al. [36], which proposed SRS for the smallest VS with a median volume of 4.06 cc and hypoFSRT for VS with a median volume of 6.71 cc. The authors demonstrated equivalent local control in both groups, with a faster shrinkage in the hypoFSRT group. There was no difference in terms of adverse events. In 2023, a meta-analysis [37] compared SRS (*n* = 353) and hypoFSRT (*n* = 511). They showed no difference in terms of tumor control, hearing preservation, facial nerve preservation, or trigeminal nerve preservation.

Persson et al. [38] carried out a review comparing SRS and normoFRST in 2017. They showed a 5.0% risk of needing remedial surgery in the SRS group vs. 4.8% in the normoFSRT group. Hearing deterioration was 49% for SRS and 45% for normoFSRT. No significant difference in nerve toxicity was found. Anderson et al. [39] published a retrospective study with three RT modalities: SRS (12.5 Gy in a single dose), normoFSRT (between 45 and 50.4 Gy), and hypoFSRT (20 Gy in five fractions). Local control was 97%, 100%, and 90.5%, respectively, for the three modalities. No difference in trigeminal toxicity or facial nerve toxicity was reported.

**Table 3 jcm-13-01611-t003:** Results of different comparative studies between SRS and FRST.

	Author	Year	Number of Patients		Technique of Radiotherapy	Median Follow-Up (Months)	Median Dose (Gy)	Median Volume (cc)	Local Control	Treatment after Radiotherapy	Toxicity			
			Total	Treatment Before							Hearing Preservation	Trigeminal Nerve Dysfunction	Facial Nerve Dysfunction	Other
COMPARISON	Huo	2020	19	NA	SRS	28.7	12.5 Gy	4.06	94.70%	NA	NA	NA	NA	31.6% radiologic oedema
			14		HypoFSRT	30.2	25 Gy in 5 fractions	6.71	92.80%	NA	NA	NA	NA	35.7% radiologic oedema
	Anderson	2014	48	24 surgeries	SRS	83.6	12.5 Gy	0.66	97% at 5 years	NA	60%	4.20%	2.1	NA
			19		normoFSRT	536	45–50.4 Gy in 25–28 fractions	2.94	100% at 5 years	NA	44.4	0.00%	0	NA
			37		hypoFSRT	43.1	20 Gy in 5 fractions	0.89	90.5% at 5 years	NA	63%	2.70%	0	NA
	Patel	2019	43	9 surgeries	SRS	26	12 Gy	diameter (17 mm)	80%	NA	70%	NA	NA	2% hydrocephalus
			57	6 surgeries	normoFSRT		50.4 Gy in 28 fractions	diameter (19 mm)	84%	NA	74%	NA	NA	2% hydrocephalus
	Udawatta	2019	21	0	SRS	31	12 Gy	diameter (13 mm)	95%	1 surgery	37.50%	NA	NA	NA
			33		normoFSRT	41	50.4 Gy in 28 fractions	diameter (18 mm)		2 surgeries	69.20%	NA	NA	NA
			6		hypoFSRT	9	25 Gy in 5 fractions	diameter (14 mm)			100%	NA	NA	NA

## 4. Discussion

Historically, RT was first used for VS as an adjuvant treatment after incomplete resection, and then SRS emerged as an alternative to microsurgery. In both SRS and FRST, the aim is to find the minimum dose to obtain the best local control and the least possible toxicity. Thus, SRS was initially performed at a dose of around 20 Gy. Significant rates of toxicity were reported, with approximately 15% facial nerve dysfunction and 15% trigeminal nerve dysfunction [13]. In the 2000s, studies showed the results of dose de-escalation to around 12 Gy and found similar local control rates and lower rates of toxicity [12].

Currently, in the absence of data from randomized controlled trials, physicians must rely upon published institutional outcomes to choose the management of vestibular schwannoma. In modern series, local control rates are consistently greater than 90% after surgical resection, FSRT, or SRS, with no significant difference between these treatment methods. In the case of KOOS 1 or KOOS 2 VS in asymptomatic patients, the wait-and-see strategy is often recommended, with MRI surveillance and audiometry at least annually, which allows seeing the rate of growth, which is a better predictor of future hearing loss than absolute tumor size [40,41,42]. Patients should be offered definitive treatment if the tumor enlarges or the hearing deteriorates.

When patients have partial hearing loss but serviceable hearing or vestibular symptoms, treatment with SRS achieves better tumor size control compared with watchful waiting with similar hearing and quality of life outcomes [43]. On the other hand, the risk of treatment-related hearing loss and other complications increases with increasing tumor size, which provides some rationale for treating early.

For patients with complete hearing loss, the goal of treatment is tumor control and preservation of nearby cranial nerve function.

Koos III VS are often symptomatic, and immediate treatment is often indicated. RT or surgery depends on the size of the cisternal component of the tumor. For KOOS 4 VS, the priority is often decompression of the brainstem and cranial nerves, so surgery is often indicated [44]. Primary RT can be reserved as a second-line treatment for a significant proportion of candidates not suitable for surgery or can be integrated into a choice of combined treatment if the residual tumor is.

No randomized trials compare the different RT approaches, and data are only available from observational studies. RT often achieves a local control rate of >90%. However, a recent comparative study [44] between SRS and microsurgery for large VS showed that long-term tumor control with SRS was inferior to surgery. As tumor volume increases, local control reduces, and the risk of edema and the need for ventricular bypass increases.

For large VS (KOOS III/KOOS IV), several prognostic factors influencing local control have been identified: a marginal dose of greater than 12 Gy as compared with less than 12 GY offers higher tumor control [22], tumor volume exceeding 10 cc [24,45], and previous microsurgery.

It is necessary to wait several years after RT to conclude that there has been a true progression and to avoid unnecessary surgical intervention. A temporary tumor expansion in 14 to 45.2% in the 2 years following RT has been classically described [46,47]. This expansion is sometimes combined with trigeminal, facial, and cochlear functional worsening [48], although the majority of symptomatic worsening was transient, and symptoms resolved after spontaneous tumor shrinkage.

Despite low complication rates with excellent tumor control, long-term hearing deterioration remains a major issue after SRS and FSRT. All of the studies [49,50,51] found SRS to be superior to surgery in preserving hearing function, with a decline to approximately 25% by 10 years [5]. However, radiation can affect neurologic function even after many years; while surgery results in immediate neurological deficit, the length of follow-up in the various studies does influence the incidence of neurologic impairment after radiation treatment.

Golfinos et al. [52] stated that SRS was superior to microsurgery in preserving hearing because of the shorter follow-up time in the SRS group. The survival curve indicated that after 60 months of postoperative follow-up, there was no difference in hearing function preservation between the SRS and microsurgery groups. This result was consistent with Carlson et al. [5] finding that the hearing function of patients after SRS will improve early postoperatively and will be decreased at the 10-year follow-up.

The etiology of this delayed hearing loss may be related to vascular insufficiency, injury to cochlear hair cells, or damage to the vestibulocochlear region itself [53,54]. Doses to the cochlea have often been studied, and the maximum and mean doses have been found to be predictive factors in various studies [55,56,57,58]:

Timmerman et al. [59] showed that doses delivered to the cochlea over 9 Gy increase the risk of hearing loss, although authors such as Chung et al. [60] reported significant hearing loss with doses greater than 6.5 Gy for SRS regimens. Bhandare et al. [61] reported 100% serviceable preservation of hearing in patients submitted to SRS for VS whose cochlea received a maximum dose less than 4.2 Gy.

For fractionated treatments, Marks et al. [62] reported hearing loss in 37% of patients who received doses greater than 60 Gy to the inner ear compared with 5% of those who received lower doses. The formal recommendation by the Quantitative Analyses of Normal Tissue Effect in the Clinic (QUANTEC) would be a median dose <45 Gy, a maximum dose <60 Gy, and the volume that receives 40 Gy corresponding to less than 2% of the volume of organ at risk.

It has also been shown [63,64] that the delay between the appearance of symptoms and the treatment is an important issue: VS patients with no subjective hearing loss at diagnosis, who were treated with SRS within 2 years of diagnosis, retained their hearing longer than patients observed and treated later.

In conclusion, the main risk associated with using radiotherapy for VS treatment is the creation of a radiation-induced tumor. Despite this risk being rare, it is critical to take it into account. Today, we know that the incidence of radiosurgery-induced tumors ranges from 0 to 3 per 200,000 patients [65].

To be recognized as a radiation-induced tumor, the lesion should follow criteria as defined by Cahan:The tumor must occur at the irradiated site;The time latency must be longer than 5 years;The tumor must be of a different pathologic type than the initial irradiated tissue.

A first review in 2010 [66] reported 14 cases of malignant VS. Among those cases, six of them were irradiated, and three had a histologic confirmation of a benign lesion before radiotherapy.

Since then, several other cases have been reported, and a recent review in 2020 [67] reported six additional cases.

In the majority of reported cases, malignant transformation led to the development of malignant peripheral nerve sheath tumors (MPNSTs) following SRS (85%). Since MPNSTs can either be radiation-induced or occur spontaneously, it cannot be definitively ruled out that at least some of the MPNSTs developed unrelated to the radiation given [68].

This risk of malignant transformation should not be a decisive factor in choosing a treatment plan, especially as the incidence of this complication is lower than the mortality rates for surgical resection. In fact, a postoperative mortality rate of 0.5% (22 out of 4886 patients) was reported in the analysis of VS treated with microsurgery in 374 hospitals in the United States from 1994 to 2003 [69], and 0.6% (25 out of 3969 patients) was reported in a review of 15 microsurgical studies [70].

Nevertheless, the clinician must be aware of the existence of this phenomenon and plan for long-term follow-up after the radiation.

## 5. Limitation

The main limitations of this review are the absence of randomized studies and the lack of long-term data, particularly for the treatment of young patients.

Finally, data on the role and outcomes of proton beam therapy are particularly lacking, and further research should be directed toward this in the management of VS.

## 6. Conclusions

SRS and FRST are non-invasive treatment options for VS. These options have been shown to successfully prevent tumor growth in >90% of patients with lesions less than 3 cm while avoiding complications associated with microsurgery. SRS is often preferred for small lesions and FSRT for larger lesions, but no randomized study between these two techniques has been carried out to confirm this approach.

## Data Availability

Data are contained within the article.

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
