# Peer review of "The Recent Management of Vestibular Schwannoma Radiotherapy: A Narrative Review of the Literature"

_jcm, 2024, doi:10.3390/jcm13061611_

Round 1

Reviewer 1 Report

Comments and Suggestions for Authors

This study is a narrative review of the literature related to radiotherapy to treat vestibular schwannomas. This is a well written review, however, the authors should discuss several points before starting with the formal review such as the natural history of tumor growth and hearing outcomes in sporadic vestibular schwannoma. The main reason is to provide a comprehensive review that will help guide the physician on how to counsel patients when offering any intervention or their vestibular schwannoma patients. Also, the most recent consensus regarding the management of sporadic vestibular schwannomas should be mentioned (Matthew CL et al 2020. Working towards consensus on sporadic vestibular schwannoma care: a modified Delphi study).

The natural history of tumor growth should be better discussed in order to inform the reader who should consider this information when counseling the patient.  The manuscript of Reznitky M, et al. (The natural history of vestibular schwannoma growth - prospective 40-year data from an unselected national cohort), is important because it talks about the real rate of tumor growth over time and the difference between intrameatal versus extrameatal tumors). Additionally, Lees KA et al. (Natural History of Sporaid Vestibular Schwnnoma: A volumetric Study of Tumor growth) looks into volume growth instead of the traditional 2D analysis. Talking about tumor growth rate in the long term is important since this will help the physician and the patient to make an adequate decision when choosing between observation vs microsurgery vs radiotherapy. This statement is also supported by the following manuscript Marinelli JP et al. (Long-term natural history and patterns of sporadic vestibular schwannoma growth: a multinstitutional volumetric analysis of 952 patients).

Regarding hearing outcomes, it is important to discuss the natural progression of hearing loss in sporadic vestibular schwannomas. I have the feeling that patients have unclear expectations regarding radiation therapy and functional hearing in the long term. The study published in 2022 from Schnurman et al. (Matched comparison of Hearing Outcomes in Patients with Vestibular Schwannoma Treated with stereotactic radio surgery or observation) is one of the most recent studies describing this. 

Another important issue to discuss is dizziness, either the development of dizziness after radiation surgery and how it compares with surgical intervention versus what to offer if the patient presents with dizziness. Most studies report that radiation does not significantly change perception of dizziness, however, surgery usually significantly improves this symptom, as reported by one of the manuscripts cited by the author written by Tatagiba M et al. (A comparative study of microsurgery and gamma knife radio surgery in vestibular schwannoma evaluating tumor control and functional outcome).

Lastly, to better compare radiation toxicity using either SRS or FSRT (HypoFSRT vs normoFSRT) authors should consider creating at table to make easy to read and compare the toxicity of different modalities.

All of this is intended to improve the comprehensive nature of this review and help the clinical to obtain the information when guiding the patient throughs the decision-making process of deciding the best treatment for their tumor. 

This is a very interesting topic with a lot of information out there. The authors have done their best to present the available data on the different modalities of radiation therapy that have been used until now.

Author Response

The suggested references have been added. Especially concerning tumor growth and long term hearing loss. 

Data about dizziness have been added.

As suggested, a new table has been added.

Reviewer 2 Report

Comments and Suggestions for Authors

In their article, the authors discuss the treatment of vestibular schwannomas with stereotactic radiosurgery and fractionated stereotactic radiotherapy. The authors thus address an interesting question.

Some aspects should be revised:

Introduction: Please introduce KOOS classification and give a brief overview of the modalities SRS and FSRT / proton RT.

The recommendation of the EANO guidelines should already be addressed here: Very large findings (>3cm) should be treated surgically (possibly combination STR with RT), smaller findings can also be irradiated

Material / Methods: Which articles were selected and why? Which were excluded and why? A flow chart would be helpful.

Results: If possible, the size of the tumours should always be taken into account

Section 3.3 - Proton radiotherapy has not yet been introduced and comes as something of a "surprise" at this point

Table 3: A separation of the four studies by lines / rows would be helpful. There is hardly any data on toxicity / aftercare, so it is questionable whether this can be listed here

Discussion: It should be emphasised again that radiotherapy is not an option for very large tumours (swelling in the course of existing compression)

Conclusion: The conclusion is very soft. What do you conclude from your analysis? When should which method be favoured?

Author Response

All suggestions were taken into account, and the new manuscript has been changed accordingly.

Reviewer 3 Report

Comments and Suggestions for Authors

The article offers a comprehensive overview of the contemporary approaches to managing vestibular schwannoma through radiotherapy.

Overall, the narrative review effectively synthesizes existing literature on the topic, providing valuable insights for clinicians and researchers. However, there are certain areas that require refinement, particularly in the description of paper selection and discussion section.

Specific comments will follow:

Specific Comments:

  1. In paragraph 3.2.2, the typographical error "lost" should be corrected to "last".
  2. It is also advisable to include information on the follow-up of the 90.5% hearing preservation rate at line 18 of the paragraph, as this adds context and completeness to the discussion.
  3.  
  4. In paragraph 4, the abbreviation "KOOS III" should be capitalized as "KOOS III" for consistency and clarity.
  5.  
  6. Additionally, the conjunction "in conclusion" may be reconsidered for appropriateness in light of subsequent content. Furthermore, the reported rate of spontaneous MPNST should be explicitly stated for completeness.

In summary, while the article offers a valuable overview of vestibular schwannoma radiotherapy management, there are opportunities for improvement in paper selection description and discussion refinement.

Nonetheless, given the scarcity of randomized trials in this field, the review provides important insights and could be accepted pending revisions to enhance clarity.

Comments on the Quality of English Language

Seems right apart for few typos.

Author Response

All suggestions have been taken into account, and the revised manuscript has been changed accordingly.